# MEDSPIKEFORMER: ALL NEURONS MATTER FOR MEDICAL IMAGE SEGMENTATION

## ABSTRACT

Spiking self-attention (SSA) has emerged as a promising approach for medical image segmentation due to its event-driven and energy-efficient nature. However, segmentation performance still degrades in complex scenarios where salient and non-salient regions coexist. Two fundamental issues remain: i) existing SSA mechanisms rely only on *activated* neurons, overlooking the contextual cues carried by *inactivated* neurons, and ii) the binary spike representation causes distribution distortions that make spiking self-attention lag behind their ANN-based self-attention (SA) in spatial discriminability. To overcome these challenges, we propose MedSpikeFormer, a spiking transformer built on the principle that *all neurons matter, both activated and inactivated*. MedSpikeFormer introduces a Spike-based Decomposed Self-Attention (SDSA) that explicitly models four types of neuronal interactions: activated–activated, activated–inactivated, inactivated–activated, and inactivated–inactivated, thus recovering rich contextual dependencies ignored by conventional SSA. Furthermore, we employ a distribution alignment loss that minimizes the divergence between SDSA and ANN-based self-attention (SA), significantly closing the performance gap to improve spatial feature discriminability while maintaining the binary nature of spiking neural networks. Extensive experiments on five medical segmentation benchmarks demonstrate that MedSpikeFormer consistently outperforms 14 state-of-the-art methods, achieving up to +2.4% mIoU on ISIC2018 and +8.7% on COVID-19. These results confirm that leveraging both fired and non-fired neurons is crucial for robust spike-driven medical image segmentation. Code is available at https://github.com/AnonymousPaper2026/MedSpikeFormer.

## 1 INTRODUCTION

Medical image segmentation (Li et al., 2024; Azad et al., 2024; Cheng et al., 2023). plays a vital role in enhancing diagnostic accuracy and assisting clinicians. The task require diverse modalities, such as dermoscopy for skin lesions (Codella et al., 2019), endoscopic polyp imaging (Jha et al., 2020), breast ultrasound (Dinh et al., 2021), nuclei microscopy (Dinh et al., 2021), and CT scans of pulmonary infections (Dong et al., 2020). Recently, spiking neural networks (SNNs) (Tavanaei et al., 2019; Shi et al., 2024; Izhikevich, 2003; Yu et al., 2025; Datta et al., 2024) have gained for their event-driven nature and low energy consumption. In SNNs, neurons transmit information via sparse binary spikes: 0 is an inactive neuron, whereas 1 denotes an activated neuron (a single firing event) (Shrestha & Orchard, 2018). While this spike-driven paradigm offers low-power benefits, it also leads to reduced segmentation accuracy in SNNs. Numerous studies (Guo et al., 2025; Huang, 2025) have integrated them with self-attention (Strudel et al., 2021; Fan et al., 2024; Bao et al., 2023) to propose spike self-attentions. However, applying them to medical image segmentation remains challenging. As shown in Figure 1, salient objects in medical images often coexist with non-salient regions, leading the model to suffer interference from non-salient areas. For instance, in dermoscopy images, lesion areas coexist with distractive elements; in endoscopy images, polyps appear alongside similar tissues. However, when the spiking self-attention (SSA) (Zhou et al., 2023; Wang et al., 2023; Yao et al., 2023; 2024) is applied to medical image segmentation, it suffers from information loss and distortion, limiting perception of both salient and non-salient regions.

**Challenge 1:** *When salient and non-salient targets coexist in medical images, how can spiking attention effectively model contextual dependencies?* Current spiking self-attention (SSA) mecha-

nisms (Zhou et al., 2023; Yao et al., 2023; Wang et al., 2023) rely solely on activated neurons while ignoring contextual cues carried by inactivated neurons, leading to severe information loss and degraded foreground–background discrimination. Although recent advances—e.g., SDSA (Yao et al., 2023; 2024) and Q-K Attention (Zhou et al., 2024)—have been proposed, they still consistently exclude inactivated neurons from the query-key dot product. As shown in Figure 1, this omission constrains contextual awareness and robustness in complex co-occurrence scenarios. Spike2Former (Lei et al., 2025) introduces integer-valued spikes and residual connections, yet still fails to fully integrate information from inactivated neurons. Consequently, it remains the core challenge to realize an efficient and comprehensive attention mechanism that encompasses all neuronal types, thereby enhancing segmentation performance in misleading co-occurrence scenes.

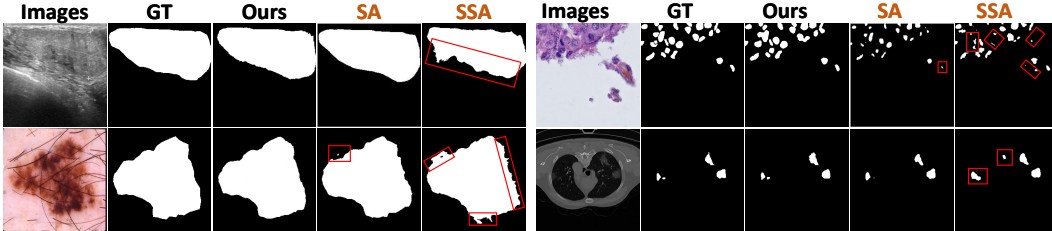

Figure 1: The misleading co-occurrence of salient and non-salient objects. SA is ANN-based self-attention and SSA denotes spike self-attention.

**Challenge 2:** *How can spike-based self-attention achieve accurate spatial-feature importance assessment, comparable to ANN-based self-attention, when salient and non-salient regions co-occur ?* Current spike self-attention (SSA) mechanisms (Zhou et al., 2023; Yao et al., 2023; Wang et al., 2023) based on Leaky Integrate-and-Fire (LIF) (Neftci et al., 2019) inherently suffer from spike–information distortion caused by their binary spike property. As illustrated in Figure 1, such distortion prevents spike self-attention from accurately evaluating the importance of spatial features in complex co-occurrence scenarios, a task at which its ANN counterpart excels. Furthermore, the study by (Qiu et al., 2025) attempts to align the information distribution between SSA and ANN-based self-attention to improve salient-object detection. However, this approach compromises the binary spike property and neglects the contextual information carried by inactivated neurons. Consequently, reducing spike-information distortion in SSA to enhance both contextual modeling capability and spatial discriminability remains an open challenge for medical image segmentation tasks.

To address the above challenges, we propose MedSpikeFormer, a spiking transformer framework. Guided by the principle that all neurons matter, we introduce the Spike-based Decomposed Self-Attention (SDSA) to address Challenge 1 by modeling all neuronal interactions. Plus, to tackle Challenge 2, we propose the distribution alignment loss to minimize the performance gap between SDSA and ANN-self attention. The contributions of this work are as follows:

- The SDSA explicitly models four types of neuronal interactions—activated–activated (A–A), activated–inactivated (A–I), inactivated–activated (I–A), and inactivated–inactivated (I–I)—to recover rich contextual dependencies.

- The distribution alignment loss minimizes the divergence between SDSA and ANN-based self-attention feature response distributions, thereby enhancing the model's discriminative ability for spatial feature importance assessment under complex co-occurrence scenarios.

- The comparative experiments and ablation studies conducted on 5 different medical image datasets demonstrate that our method outperforms 14 state-of-the-art methods.

## 2 PRELIMINARIES

This section introduces Spiking Neural Networks (SNNs) and spike-based attention mechanisms.

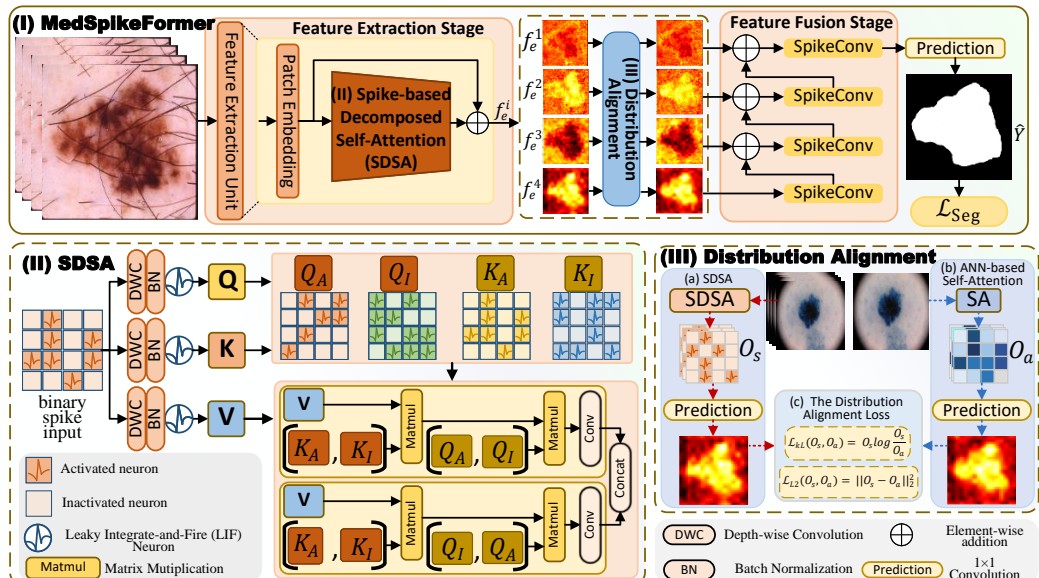

Figure 2: Illustration of the overall architecture of MedSpikeFormer.

## 2.1 SPIKING NEURON MODELS

**Leaky Integrate-and-Fire neuron (LIF).** The LIF neuron (Maass, 1997; Luo et al., 2024; Guo et al., 2024) models membrane potential dynamics using temporal accumulation and thresholding:

$$U[t] = H[t-1] + X[t] \tag{1}$$

$$S[t] = \Theta(U[t] - V_{th}) \tag{2}$$

$$H[t] = \beta(U[t] - S[t]) \tag{3}$$

where $U[t]$ is the membrane potential, $X[t]$ is the input at timestep $t$, $H[t-1]$ is the prior state, $\beta$ is the decay factor, and $\Theta(\cdot)$ is the Heaviside step function, with the firing threshold $V_{th}$. It generates binary spikes $S[t] \in \{0, 1\}$, whose coarse quantization limits the neuron's ability to represent nuanced spatial information—an issue critical for pixel-level medical image segmentation tasks.

## 2.2 SELF-ATTENTION MECHANISMS

**ANN-based Self-attention.** Given an input sequence $\mathbf{X} \in \mathbb{R}^{N \times C}$, where $N$ is the number of tokens (e.g., spatial positions) and $C$ is the feature dimension (e.g., channels), ANN-based self-attention (Dosovitskiy et al., 2021) generates float-form query $Q$, key $K$, and value $V$ matrices as:

$$Q, K, V = \mathbf{X}W_{\mathcal{Q}}, \mathbf{X}W_{\mathcal{K}}, \mathbf{X}W_{\mathcal{V}} \tag{4}$$

where $W_{\mathcal{Q}}, W_{\mathcal{K}}, W_{\mathcal{V}} \in \mathbb{R}^{C \times C}$ are learnable weights. The output attention is computed via:

$$\mathcal{O}_a = \text{SA}(Q, K, V) = \text{Softmax}(\frac{Q \otimes K^T}{\sqrt{C}}) \otimes V \tag{5}$$

where $\otimes$ denotes a matrix multiplication operation.

**Spike Self-attention.** Given an input sequence $\mathbf{X} \in \mathbb{R}^{N \times C}$, spike self-attention (SSA) (Zhou et al., 2023) generates spike-form query $Q$, key $K$, and value $V \in \{0, 1\}$ as:

$$Q, K, V = \text{SNN}(\mathbf{X}W_{\mathcal{Q}}, \mathbf{X}W_{\mathcal{K}}, \mathbf{X}W_{\mathcal{V}}) \tag{6}$$

where $SNN(\cdot)$ is spike activation function chosen as LIF. The output attention is computed via:

$$\mathcal{O}_s = \text{SSA}(Q, K, V) = \left(\frac{Q \otimes K^{\top}}{\sqrt{C}}\right) \otimes V \tag{7}$$

where $Q, K, V \in \{0, 1\}^{N \times C}$ are binary spike matrices. Moreover, the attention is restricted to pairs where both neurons are activated, ignoring inactivated entries and thus limiting contextual modeling in challenging segmentation contexts.

## 3 METHOD

### 3.1 PROBLEM SETUP

Medical image segmentation using Spiking Neural Networks (SNNs) faces two fundamental challenges: the *Sparse Attention Problem* and the *Spike-information Distortion Problem*: the first arises when spike-based attention mechanisms compute interactions only among activated neurons, yielding a sparse attention support set:

$$\mathcal{S}_{\text{SA}} = \{(i,j) \,|\, Q_{i,j} > 0 \wedge K_{i,j} > 0\}, \quad |\mathcal{S}_{\text{SA}}| \ll N^2,$$

restricting context modeling, and while the second stems from the binary spike characteristic of conventional spiking neurons, which causes the spiking Transformer to lag behind its ANN-based Transformer in segmentation performance:

$$D_{KL}(\mathcal{O}_s || \mathcal{O}_a) \geq \epsilon_1, \quad ||\mathcal{O}_s - \mathcal{O}_a||_2^2 \geq \epsilon_2, \quad \epsilon_1 \gg 0, \epsilon_2 \gg 0,$$

leading to the degraded spatial discrimination, which is due to the information distribution misalignment between the spiking Transformer and ANN-based Transformer.

Together, these challenges lead to the following constrained optimization objective:

$$\min_{\theta} \mathcal{L}_{\text{seg}}(f_{\theta}(\mathbf{X}), \mathbf{Y}) \text{ s.t. } \mathcal{S}_{\text{enh}} \supseteq \mathcal{S}_{\text{SA}}, \ D_{KL}(\mathcal{O}_s || \mathcal{O}_a) + ||\mathcal{O}_s - \mathcal{O}_a||_2^2 \approx 0,$$

where $\theta \in \mathbb{R}^d$ denotes the learnable network parameters, $\mathbf{X} \in \mathbb{R}^{N \times C}$ is the input feature map, $\mathbf{Y} \in \mathbb{R}^{N \times 1}$ is the corresponding ground-truth segmentation, $\mathcal{S}_{\text{SA}} \subset \{1, \ldots, N\}^2$ is the sparse attention support set computed from activated neuron pairs, $\mathcal{S}_{\text{enh}}$ denotes the enhanced support set with broader spike interactions with $\mathcal{S}_{\text{enh}} \supseteq \mathcal{S}_{\text{SA}}$, $D_{KL}(\cdot)$ denotes the degree of information-distribution mismatch between $O_s$ and $O_a$; $|| \cdot ||_2^2$ denotes the pixel-level discrepancy between $O_s$ and $O_a$.

Building on this setup, we design MedSpikeFormer with the key modules: Spike-based Decomposed Self-Attention (SDSA). First, the SDSA expands spike interaction coverage by explicitly modeling all neuron pair types—activated–activated (A–A), activated–inactivated (A–I), inactivated–activated (I–A), and inactivated–inactivated (I–I)—thereby mitigating the sparse attention problem. Second, the distribution alignment losses ($D_{KL}(\cdot)$ and $|| \cdot ||_2^2$) are introduced to minimize the divergence between SDSA and ANN-based self-attention feature response distributions, thereby effectively resolving the spike-information distortion. To this end, these modules are integrated into a multi-stage encoder–decoder framework for robust spatial encoding and segmentation under spike constraints.

### 3.2 SDSA: EXPANDING SPIKE ATTENTION SCOPE

Salient object detection in medical image segmentation often suffers from misleading co-occurrence between salient and non-salient regions, as shown in Figure 1. Traditional spike-based attention mechanisms (Zhou et al., 2023; 2024) compute attention maps only among activated neurons, discarding contextual cues from inactivated neurons. This sparse representation limits the model's ability to perceive complete spatial information and degrades performance in complex scenes.

To address this, we propose the Spike-based Decomposed Self-Attention (SDSA), which expands the attention scope by explicitly modeling all possible neuron pair interactions — A-A, A-I, I-A and I-I —thus preserving richer information flows across the network.

**Definition 1. Complementary Spike Decomposition.** *Given discrete binary spike-form query and key matrices* $Q, K \in \{0, 1\}^{N \times C}$ *generated by LIF neurons, we define their* activated *and* complementary inactivated *components as:*

$$Q_{\mathcal{A}} = Q, \quad Q_{\mathcal{I}} = 1 - Q, \quad K_{\mathcal{A}} = K, \quad K_{\mathcal{I}} = 1 - K,$$

*where* $Q_{\mathcal{A}}, K_{\mathcal{A}}$ *capture neuron activations, while* $Q_{\mathcal{I}}, K_{\mathcal{I}}$ *represent complementary responses from inactivated neurons.*

This decomposition enables complementary modeling of spike activations and their inactivated counterparts, which is crucial for attention mechanisms operating on binary spikes. By considering both $Q_{\mathcal{A}}, K_{\mathcal{A}}$ and $Q_{\mathcal{I}}, K_{\mathcal{I}}$, SDSA constructs an enriched attention support that alleviates the sparsity introduced by traditional spike attention, ensuring that information from all spiking units — activated or inactivated — contributes to the segmentation process.

**Proposition 1. Information Gain of SDSA.** *Let $S_{SA}$ denote the output of conventional spike attention (e.g., SSA, Q-K attention) which only models interactions among activated neurons. Suppose the inactivated components ($Q_{\mathcal{I}} = 1$ or $K_{\mathcal{I}} = 1$) carry non-trivial information about the input $X$. Then the enhanced attention output $S_{enh}$ constructed by SDSA satisfies:*

$$I(X; S_{enh}) = I(X; Q_{\mathcal{A}}, Q_{\mathcal{I}}, K_{\mathcal{A}}, K_{\mathcal{I}}) > I(X; Q_{\mathcal{A}}, K_{\mathcal{A}}) = I(X; S_{SA}).$$

*where $I(\cdot\,;\cdot)$ denotes the mutual information.*

Proof can be found in the Section B of Supplementary Material. Conventional spike attention restricts the attention support to pairs $(i, j)$ where both $Q_{i,j} = 1$ and $K_{i,j} = 1$, yielding:

$$S_{SA} = \phi(Q_{\mathcal{A}}, K_{\mathcal{A}}),$$

where $\phi$ denotes a simplified token-level interaction such as element-wise multiplication. According to Definition 1, any pair involving inactivated components ($Q_{\mathcal{I}} = 1$ or $K_{\mathcal{I}} = 1$) is excluded, limiting information flow and representation capacity, especially in cases where semantic signals lie in inactivated neurons (e.g., blurred lesions, fuzzy boundaries).

SDSA overcomes this by incorporating the full interaction space through spike decomposition. Specifically, we compute spike-level attention weights as:

$$S_{enh} = \Big[ \big[Q_{\mathcal{A}}, Q_{\mathcal{I}}\big] \otimes \big[K_{\mathcal{A}}, K_{\mathcal{I}}\big]^T ; \big[Q_{\mathcal{I}}, Q_{\mathcal{A}}\big] \otimes \big[K_{\mathcal{A}}, K_{\mathcal{I}}\big]^T \Big] \tag{8}$$

These decomposed attention weights are contributed to $V$ via matrix multiplication to yield:

$$\mathcal{O}_s = \Big[ \frac{\big[Q_{\mathcal{A}}, Q_{\mathcal{I}}\big] \otimes \big[K_{\mathcal{A}}, K_{\mathcal{I}}\big]^T}{\sqrt{C}} \otimes V ; \frac{\big[Q_{\mathcal{I}}, Q_{\mathcal{A}}\big] \otimes \big[K_{\mathcal{A}}, K_{\mathcal{I}}\big]^T}{\sqrt{C}} \otimes V \Big] \tag{9}$$

which explicitly models all interactions: A-A, A-I, I-A, and I-I. Plus, we rigorously ablate the contribution of each interaction type (see Table 3 in Section 4.3), confirming that all interactions are critical for segmentation performance.

This proposition theoretically supports SDSA's ability to enhance information retention in spiking attention. By reintroducing inactivated neurons into the attention computation, SDSA is better suited for dense prediction tasks like medical image segmentation where clinically relevant details may reside in ambiguous, low-saliency regions. As shown in Figure 1, this mechanism leads to improved contextual modeling and segmentation performance in complex co-occurrence scenarios.

## 3.3 DISTRIBUTION ALIGNMENT BETWEEN SDSA AND ANN-BASED SELF-ATTENTION

Motivated by the spike-information distortion in conventional spiking attention—where binary spike properties cause significant divergence in attention distributions between Spiking self-attention and ANN-based self-attention—we address this misalignment as evidenced by Figure 1: SDSA's attention focus differs markedly from ANN-based self-attention, with ANN-based self-attention demonstrating superior lesion localization accuracy. This distribution mismatch impairs spatial discriminability under complex co-occurrence scenarios. To resolve this, we propose a distribution alignment loss that minimizes the divergence between SDSA and ANN-based self-attention feature response distributions, thereby correcting SDSA's attention focus for precise medical segmentation while preserving the binary nature of spike neurons.

**Definition 2. Distribution alignment.** *Given the SDSA's attention output $\mathcal{O}_s$ and the ANN-based self-attention's attention output $\mathcal{O}_a$, as shown in Figure 2, they are transformed to $\hat{\mathcal{O}}_s$ and $\hat{\mathcal{O}}_a$ through learnable linear layers that map features to pixel-level segmentation logits. $\hat{\mathcal{O}}_s$ and $\hat{\mathcal{O}}_a$ are the feature response distributions of SDSA and ANN-based self-attention respectively. The distribution alignment between them is formulated as a distribution alignment loss, as follows:*

$$\mathcal{L}_{align} = \frac{1}{N} \sum_{(i,j) \in \{1,\ldots,N\}^2} \left[ \hat{\mathcal{O}}_s^{i,j} \log\left( \frac{\hat{\mathcal{O}}_s^{i,j}}{\hat{\mathcal{O}}_a^{i,j} + \epsilon} \right) + \left\| \hat{\mathcal{O}}_s^{i,j} - \hat{\mathcal{O}}_a^{i,j} \right\|_2^2 \right]$$

*where $\hat{\mathcal{O}}_{s/a}^{(i,j)}$ denotes the feature value at position $(i, j)$ in $\hat{\mathcal{O}}_{s/a}$, with $(i, j) \in \{1, \ldots, N\}^2$. The term $\hat{\mathcal{O}}_s \log(\frac{\hat{\mathcal{O}}_s}{\hat{\mathcal{O}}_a + \epsilon})$ minimizes information distribution mismatch in attention focus, and the term $\|\hat{\mathcal{O}}_s - \hat{\mathcal{O}}_a\|_2^2$ enforces pixel-level spatial discriminability.*

This formulation addresses the spike-information distortion problem by minimizing the discrepancy between the attention distributions of SDSA and ANN-based self-attention, enabling precise medical image segmentation while preserving spike sparsity, as further analyzed in Proposition 2.

**Proposition 2. Gradient Dynamics and Stability of Distribution Alignment.** *Under the standard assumption that $\mathcal{O}_s \in [\epsilon, 1]$ and $\mathcal{O}_a \in [\epsilon, 1]$ for a small $\epsilon > 0$ (ensuring no log-domain singularity), the combined gradient effectively guides the optimization process to address the aforementioned challenges in practice:*

$$\nabla \mathcal{L}_{align} = \frac{\partial \mathcal{L}_{align}}{\partial \mathcal{O}_s} = \frac{1}{N} \left( \log \frac{\mathcal{O}_s}{\mathcal{O}_a} + 1 \right) + \frac{2}{N} (\mathcal{O}_s - \mathcal{O}_a),$$

where the gradient $\frac{1}{N}(\log \frac{\mathcal{O}_s}{\mathcal{O}_a} + 1)$ governs distributional alignment by measuring relative discrepancies in probabilistic structure. The gradient $\frac{2}{N}(\mathcal{O}_s - \mathcal{O}_a)$ enforces geometric fidelity by penalizing absolute pixel-wise errors.

*Further, the Frobenius norm of the gradient satisfies:*

$$||\nabla \mathcal{L}_{align}||_F \leq \frac{1}{N} \sqrt{N^2 (\log \epsilon + 1)^2} + \frac{2}{N} \sqrt{N^2 (1 - \epsilon)^2} = 3 - 2\epsilon - \log \epsilon,$$

This guarantees that gradients remain bounded for any matrix, which ensures convergence. Therefore, the $\nabla \mathcal{L}_{align}$ effectively minimizes the divergence between SDSA and ANN-based self-attention feature response distributions, thereby reducing spike-information distortion to enhance segmentation performance under cluttered or low-contrast conditions. Proof can be found in the Section C of Supplementary Material.

## 3.4 OVERALL ORCHESTRATION OF MEDSPIKEFORMER

MedSpikeFormer is designed with a two-stage architecture to progressively extract and integrate hierarchical features for accurate medical image segmentation. As illustrated in Figure 2, the two main stages are: *Stage I—Feature Extraction* and *Stage II—Feature Fusion*. Pseudocode and detailed description are provided in Section D of the Supplementary Material.

**Stage I: Feature Extraction.** This stage contains three cascaded encoder units. Each unit has a Patch Embedding (PE) module followed by the proposed SDSA module. The PE module integrates a convolution layer and a spike convolution to encode local spatial patterns, while SDSA enhances global interaction via complementary spike decomposition. The output of the $i$-th encoder unit is:

$$f_e^i = \text{SDSA}(\text{PE}(X)) + \text{PE}(X), \tag{10}$$

where $X \in \mathbb{R}^{T \times H \times W \times C}$ is the input feature map, and $f_e^i \in \mathbb{R}^{T \times \frac{H}{2^{i-1}} \times \frac{W}{2^{i-1}} \times 2^{i-1}C}$ $(i \in \{1, 2, 3, 4\})$ is the output after feature enhancement and residual connection.

**Stage II: Feature Fusion.** This stage performs progressive integration of features extracted at different depths using spike convolution blocks. Deep semantic features are refined, and shallow information is preserved through skip connections. The fusion process is formulated as:

$$f_s^i = \begin{cases} \text{SpikeConv}(f_e^i), & i = 4, \\ \text{SpikeConv}(f_e^i + f_s^{i+1}), & i \in \{1, 2, 3\}, \end{cases} \tag{11}$$

where $f_s^i$ is the fused output at $i$-th layer, and $\text{SpikeConv}(\cdot)$ denotes the spike convolution operation with discretized binary outputs.

**Prediction Head.** The final prediction $\hat{Y}$ is generated from the shallowest fused feature $f_s^1$ via a prediction head:

$$\hat{Y} = \phi_s(f_s^1, \omega_s), \tag{12}$$

where $\phi_s(\cdot)$ is a learnable transformation with parameters $\omega_s$ that maps features to pixel-level segmentation logits.

**Loss Function.** To optimize MedSpikeFormer for accurate medical image segmentation, we adopt a composite objective function that balances region-level and pixel-level supervision. Specifically, we define the segmentation loss as:

$$\mathcal{L}_{seg} = \underbrace{-\sum_i \left[ Y_i \log(\hat{Y}_i) + (1 - Y_i) \log(1 - \hat{Y}_i) \right]}_{\mathcal{L}_{BCE}} + \underbrace{1 - \frac{2 \sum_i \hat{Y}_i Y_i + \epsilon}{\sum_i \hat{Y}_i^2 + \sum_i Y_i^2 + \epsilon}}_{\mathcal{L}_{Dice}}, \quad (13)$$

where $Y_i \in \{0, 1\}$ is the ground truth label for the $i$-th pixel, and $\hat{Y}_i \in [0, 1]$ is the predicted mask value. The term $\epsilon$ is a small constant for numerical stability. $\mathcal{L}_{BCE}$ provides fine-grained pixel-wise guidance and $\mathcal{L}_{Dice}$ encourages global structure alignment. The total loss function is defined as:

$$\mathcal{L}_{total} = \mathcal{L}_{seg} + \mathcal{L}_{align}, \quad (14)$$

*Objective Alignment.* This loss design reflects the core motivation of MedSpikeFormer: to enable spike-driven segmentation networks to retain global contextual cues while preserving fine-grained boundaries and textures. Specifically, $\mathcal{L}_{seg}$ emphasizes shape integrity and region-wise coverage and focuses on detail preservation and sharps boundaries, aligning with the goal of mitigating fragmented predictions in complex medical imaging scenarios. $\mathcal{L}_{align}$ is essential for recovering spatial precision lost to reduce the spike-information distortion. By jointly optimizing these two components, $\mathcal{L}_{total}$ enables the model to robustly learn salient object distributions under spike constraints—bridging the global-local gap induced by event-driven computation.

## 4 EXPERIMENT

### 4.1 DATASETS AND EXPERIMENT DETAILS

Our method is evaluated on 5 medical datasets, including the ISIC2018 (Codella et al., 2019), Kvasir (Jha et al., 2020), BUSI (Al-Dhabyani et al., 2020), Moun-Seg (Dinh et al., 2021), and COVID-19 (Dong et al., 2020). Plus, we adopt 5 metrics (Ruan et al., 2024), including Mean Intersection over Union (mIoU), Dice Similarity Coefficient (DSC) , Accuracy (Acc), Specificity (Spe) and Sensitivity (Sen). For detailed information on datasets and experimental setups, refer to the Section E.1 of Supplementary Material.

### 4.2 COMPARISON WITH STATE-OF-THE-ARTS

We compare our method with 14 current methods on 5 medical image datasets, and meanwhile, all models are trained and evaluated under the same experiment settings.

**Quantitative Comparison.** Table 1 shows the superior segmentation performance of our model. Specifically, in the mIoU metric, on the ISIC2018 and Kvasir datasets, our method improves STDv3 by 3.47% and 2.72%, respectively. Notably, on the COVID-19 dataset, our method improves the MFMSA by 8.06%. on the Monu-Seg dataset, our model improves the UCTransNet by 1.78%, showing the robustness of our method in the multi-object scenario. Plus, on the BUSI dataset, our method achieves the best segmentation performance, indicating the robustness of our model in low-contrast scenes. See Section E.2 and E.3 of Supplementary Material for statistical significance tests, mIoU curves and more details.

**Visual Comparison.** As shown rows 3 and 5 in Figure 3, in scenes of low-contrast and fuzzy edges in medical images, other methods perform poorly in salient object edge detection and are easily affected. However, our method can effectively identifies the segmentation boundaries of salient objects. Plus, as shown rows 3 and 4 in Figure 3, in multi-object detection scenarios, especially when there are a large number of objects and the object size is small, other methods are easily affected by the background. However, our method performs well in this case. Experimental results show the superior capability of our method in image segmentation tasks. See Figure 1 of Supplementary Material for more visualization comparisons.

**Computational Efficiency.** Though not lightweight-oriented, MedSpikeFormer keeps low overhead with 2.9M parameters and 26.87 GFLOPs. It outperforms larger models such as SDSA (13.6M, 28.6 GFLOPs) and FSTA-SNN (45.7M, 20.6 GFLOPs), while preserving spiking sparsity. Plus, hardware-friendly energy efficiency are detailed in Section E.4 of the Supplementary Material.

Table 1: Performance comparison with 14 SOTA methods on 5 medical image datasets.

| Dataset | Metric | U-Net MICCAI 2015 | UCTransNet AAAI 2022 | D-LKA WACV 2023 | EGE-UNet MICCAI 2023 | SAM-Med2D arXiv 2023 | SDSA NeurIPS 2023 | MLW-Net CVPR 2023 | UltraLight VM-UNet arXiv 2024 | MFMSA CVPR 2024 | VPTTA CVPR 2024 | EMCAD CVPR 2024 | QKFormer NeurIPS 2024 | STDV3 TPAMI 2025 | FSTA-SNN AAAI 2025 | Our 2025 |
|---|---|---|---|---|---|---|---|---|---|---|---|---|---|---|---|---|
| ISIC 2018 | mIoU↑ | 0.8004 | 0.8185 | 0.8033 | 0.8108 | 0.7383 | 0.7853 | 0.7650 | 0.8110 | 0.8163 | 0.7842 | 0.8071 | 0.7706 | 0.8303 | 0.6403 | 0.8550 |
| | DSC↑ | 0.8891 | 0.9002 | 0.8909 | 0.8955 | 0.8494 | 0.8579 | 0.8613 | 0.8956 | 0.8988 | 0.8790 | 0.8932 | 0.8614 | 0.8965 | 0.7400 | 0.9081 |
| | Acc↑ | 0.9513 | 0.9565 | 0.9514 | 0.9535 | 0.9397 | 0.9572 | 0.9404 | 0.9527 | 0.9549 | 0.9467 | 0.9531 | 0.9475 | 0.9666 | 0.9110 | 0.9812 |
| Kvasir | mIoU↑ | 0.7330 | 0.7670 | 0.7212 | 0.5604 | 0.5911 | 0.7970 | 0.6636 | 0.6100 | 0.7625 | 0.5164 | 0.7173 | 0.7658 | 0.8215 | 0.6313 | 0.8534 |
| | DSC↑ | 0.8459 | 0.8681 | 0.8380 | 0.7182 | 0.7430 | 0.8726 | 0.7977 | 0.7577 | 0.8652 | 0.6811 | 0.8353 | 0.8521 | 0.8915 | 0.7642 | 0.9093 |
| | Acc↑ | 0.9253 | 0.9579 | 0.9495 | 0.9108 | 0.9324 | 0.9741 | 0.9378 | 0.9190 | 0.9568 | 0.9076 | 0.9478 | 0.9662 | 0.9722 | 0.9264 | 0.9804 |
| Moun-Seg | mIoU↑ | 0.6784 | 0.6890 | 0.6300 | 0.5009 | 0.2699 | 0.6554 | 0.6535 | 0.5600 | 0.6111 | 0.4151 | 0.5603 | 0.6003 | 0.6696 | 0.6495 | 0.7014 |
| | DSC↑ | 0.8084 | 0.8159 | 0.7730 | 0.6674 | 0.4250 | 0.7883 | 0.7904 | 0.7180 | 0.7586 | 0.5867 | 0.7182 | 0.7475 | 0.8008 | 0.5834 | 0.8238 |
| | Acc↑ | 0.9348 | 0.9433 | 0.9284 | 0.8873 | 0.9433 | 0.9414 | 0.9332 | 0.9094 | 0.9338 | 0.8662 | 0.9263 | 0.9335 | 0.9467 | 0.9095 | 0.9518 |
| COVID-19 | mIoU↑ | 0.3605 | 0.3971 | 0.3098 | 0.3912 | 0.4025 | 0.5062 | 0.4295 | 0.5532 | 0.6262 | 0.4591 | 0.4120 | 0.4701 | 0.5974 | 0.4902 | 0.7138 |
| | DSC↑ | 0.5300 | 0.5684 | 0.4730 | 0.5624 | 0.5739 | 0.6261 | 0.6009 | 0.7123 | 0.7201 | 0.6293 | 0.5835 | 0.6357 | 0.6933 | 0.6517 | 0.8009 |
| | Acc↑ | 0.9784 | 0.9804 | 0.9766 | 0.9805 | 0.9856 | 0.9782 | 0.9808 | 0.9867 | 0.9901 | 0.9859 | 0.9790 | 0.9715 | 0.9286 | 0.9713 | 0.9565 |
| BUSI | mIoU↑ | 0.4775 | 0.5870 | 0.4969 | 0.5103 | 0.4770 | 0.5115 | 0.4811 | 0.4743 | 0.5771 | 0.4420 | 0.4620 | 0.4310 | 0.2854 | 0.5582 | 0.6006 |
| | DSC↑ | 0.6463 | 0.7382 | 0.6639 | 0.6758 | 0.6459 | 0.5917 | 0.6496 | 0.6434 | 0.7318 | 0.6131 | 0.6320 | 0.6041 | 0.3774 | 0.7165 | 0.6676 |
| | Acc↑ | 0.9605 | 0.9637 | 0.9502 | 0.9551 | 0.9616 | 0.9571 | 0.9462 | 0.9491 | 0.9629 | 0.9426 | 0.9412 | 0.9377 | 0.9362 | 0.9652 | 0.9759 |
| Params (M) ↓ | | 14.7518 | 66.2424 | 22.8401 | 0.0458 | - | 13.5588 | 94.9794 | 0.0376 | 31.2192 | 22.0224 | 26.7643 | 16.9599 | 25.5286 | 45.6777 | 1.7369 |
| GFLOPs ↓ | | 32.8948 | 30.9839 | 16.8894 | 0.0072 | - | 28.5805 | 108.0758 | 0.0602 | 9.9752 | 40.0514 | 5.5960 | 32.8858 | 12.3349 | 20.6515 | 19.0929 |

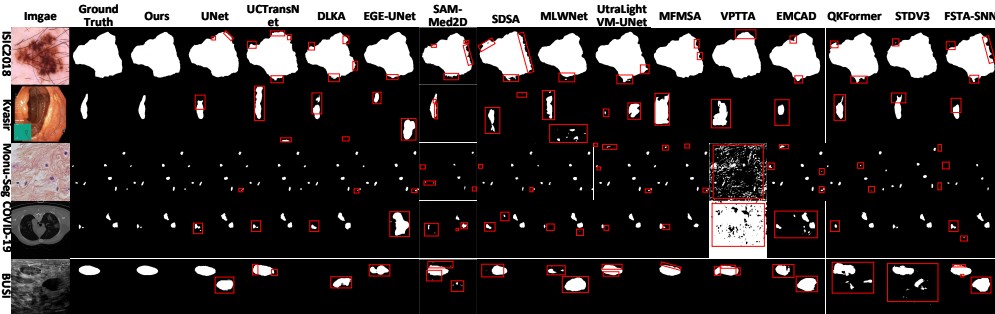

Figure 3: We compare our method with 14 methods. The red box denotes incorrect prediction areas.

## 4.3 ABLATION STUDIES

To thoroughly validate the design of the MedSpikeFormer, we conduct extensive ablation experiments on 5 datasets using the mIoU metric. Our investigations center around 6 key questions:

Q1: How critical is the Spike-based Decomposed Self-Attention (SDSA)? Removing SDSA (w/o SDSA) leads to drops in mIoU, e.g., $-9.07\%$ on ISIC2018, as shown in Table 2, showing SDSA's importance for capturing comprehensive attention in complex co-occurrence scenarios. Plus, Figure 4 shows that removing SDSA (w/o SDSA) leads to drops in segmentation performance. Then, we further provide heatmaps to show SDSA's effectiveness (Figure 5). See Section E.5 and Figure 3 in the Supplementary Material for more details.

Table 2: Ablation on SDSA, ANN-based self-attention and Spike Conv (mIoU).

| Model Variant | ISIC2018 | Kvasir | Monu-Seg | COVID-19 | BUSI | Params (M) | GFLOPs |
|---|---|---|---|---|---|---|---|
| Ours | **0.8550** | **0.8534** | **0.7014** | **0.7138** | **0.6006** | 2.8745 | 26.8972 |
| w/o SDSA | 0.7727 | 0.7866 | 0.4785 | 0.3869 | 0.5231 | 2.1258 | 15.2046 |
| w/o SA | 0.7866 | 0.7647 | 0.5462 | 0.5792 | 0.5384 | 2.8745 | 26.8972 |
| w/o SC | 0.7999 | 0.8096 | 0.6166 | 0.6049 | 0.5508 | 2.3328 | 20.8422 |

Q2: Does the ANN-based self-attention (SA) module significantly affect performance? Yes. Removing ANN-based self-attention (w/o SA) leads to notable performance degradation across all datasets (e.g., $-8.87\%$ on Kvasir), as shown in Table 2, confirming its effectiveness. Plus, Figure 4 also shows that removing ANN-based self-attention (w/o SA) descends the segmentation performance. See Section E.5 and Figure 3 in the Supplementary Material for visualization comparisons.

Q3: Does the Spike Convolution (SC) module improve performance? Yes. Removing Spike Conv (w/o SC) leads to notable performance degradation across all datasets (e.g., $-4.98\%$ on BUSI), as shown in Table 2, confirming its effectiveness for segmentation performance. Similarly, as shown in Figure 4, removing the Spike Convolution (w/o SC) leads to performance degradation.

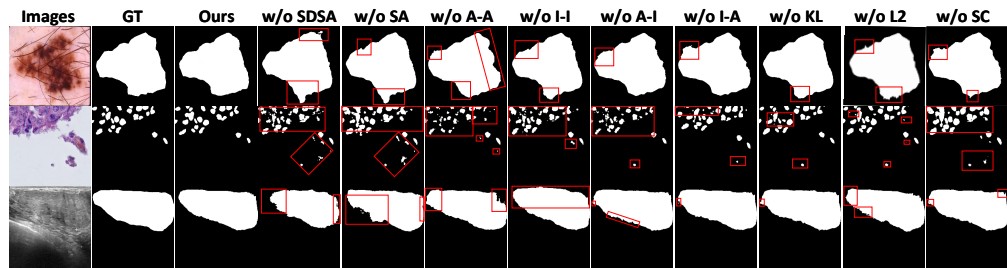

Figure 4: Visual Ablation Comparison. The red box indicates the area of incorrect predictions.

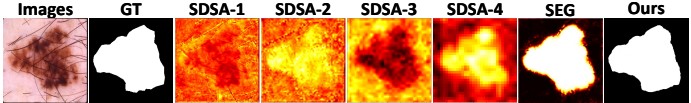

Figure 5: Visual Ablation Heatmaps.

Table 3: Ablation on interaction types in SDSA (mIoU).

| Model Variant | ISIC2018 | Kvasir | Monu-Seg | COVID-19 | BUSI |
|---|---|---|---|---|---|
| Ours | **0.8550** | **0.8534** | **0.7014** | **0.7138** | **0.6006** |
| w/o A-A | 0.8243 | 0.8054 | 0.6814 | 0.6411 | 0.5281 |
| w/o A-I | 0.8567 | 0.8116 | 0.7063 | 0.6895 | 0.5673 |
| w/o I-A | 0.8492 | 0.8282 | 0.7079 | 0.6763 | 0.5585 |
| w/o I-I | 0.8344 | 0.8140 | 0.6988 | 0.6519 | 0.5032 |
| w/o KL | 0.7743 | 0.7782 | 0.4988 | 0.3988 | 0.5108 |
| w/o $L_2$ | 0.7743 | 0.7782 | 0.4988 | 0.3988 | 0.5108 |

Q4: Do all four types of spike interactions in SDSA matter? Yes. Table 3 shows that removing any of the four interaction types (w/o A-A, w/o A-I, w/o I-A, and w/o I-I) leads to performance drop, confirming that all interaction pairs contribute to comprehensive information modeling. Plus, Figure 4 shows that removing any of the four interaction types leads to performance drop. See Section E.5 and Figure 3 in the Supplementary Material for visualization comparisons and more details.

Q5: Does the distribution alignment loss improve segmentation performance? Yes. The loss consists of KL divergence and $L_2$ loss. As shown in Table 3, removing either loss component (w/o KL or w/o $L_2$) leads to a drop in segmentation performance, with the $L_2$ loss having a particularly significant impact. Plus, Figure 4 also confirms the effectiveness of the distribution alignment loss.

Plus, we also conduct following ablation studies: Q6: Is the timestep $D$ in our method important? For more details of the ablation study, refer to Section E.5 of the Supplementary Material.

## 5 OBSERVATIONS AND LIMITATIONS

As shown in Figure 3, MedSpikeFormer can effectively identify salient objects in scenarios with blurred boundaries and multiple objects, but subtle deviations from the ground truth still exist. Future work will explore efficient edge detection techniques, thereby further enhancing overall segmentation performance. For more details refer to Section F of Supplementary Material.

## 6 CONCLUSION

This paper presents the MedSpikeFormer, which significantly improves segmentation performance in scenarios with the co-occurrence of salient and non-salient objects via the collaborative work of the Spike-based Decomposed Self-attention and the distribution alignment loss. Experimental results show that MedSpikeFormer outperforms 14 methods across 5 public medical image datasets, underscoring its strong capability in medical image segmentation. Potentially, we will further explore efficient spike-driven edge detection methods to enhance segmentation performance.

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
