# MEDSPIKEFORMER

## A CODE LINK

Our code is available on https://github.com/AnonymousPaper2026/MedSpikeFormer.

## B PROOF OF THE PROPOSITION 1

**Proposition 1. Information Gain of SDSA.** *Let $S_{SA}$ denote the output of conventional spike atten- tion (e.g., SSA, Q-K attention) which only models interactions among activated neurons. Suppose the inactivated components ($Q_{\mathcal{I}} > 0$ or $K_{\mathcal{I}} > 0$) carry non-trivial information about the input $X$. Then the enhanced attention output $S_{enh}$ constructed by SDSA reduces information loss compared to traditional spike attention, and satisfies:*

$$H(X|S_{enh}) < H(X|S_{SA})$$

*Equivalently in mutual information form:*

$$\begin{aligned} I(X; S_{enh}) &= I(X; Q_{\mathcal{A}}, Q_{\mathcal{I}}, K_{\mathcal{A}}, K_{\mathcal{I}}) \\ &> I(X; Q_{\mathcal{A}}, K_{\mathcal{A}}) = I(X; S_{SA}). \end{aligned}$$

*where $I(\cdot\,;\cdot)$ denotes the mutual information.*

**Proof 1.** The proof consists of three parts establishing strict information gain from inactivated neurons.

*Part 1: Information Loss in Traditional Approach.* For any inactivated neuron $(i, j)$ where $Q^{ij} = 0$:

$$H(X_{ij}|Q^{ij} = 0) = H(X_{ij})$$
$$I(X_{ij}; Q^{ij}) = 0$$

since $Q^{ij} = 0$ provides no information about sub-threshold values $X_{ij} \in (0, \theta)$. The traditional output $S_{SA}$ discards all sub-threshold information:

$$H(X|S_{SA}) = H(X|Q_{\mathcal{A}}, K_{\mathcal{A}}) \geq \sum_{\substack{i,j \\ Q^{ij}=0}} H(X_{ij})$$

*Part 2: Information Preservation in Decomposition.* For inactivated neurons, $Q_I^{ij} = 1$ when $Q^{ij} = 0$. This preserves knowledge of sub-threshold activation:

$$\begin{aligned} I(X_{ij}; Q_{\mathcal{I}}^{ij}) &= H(Q_{\mathcal{I}}^{ij}) - H(Q_{\mathcal{I}}^{ij}|X_{ij}) \\ &= H_b(p_0) - (1 - p_0)H_b\left(\frac{p_\theta}{1 - p_0}\right) > 0 \end{aligned}$$

where $p_0 = P(X_{ij} < \theta)$, $p_\theta = P(0 < X_{ij} < \theta)$, and $H_b$ is binary entropy. The strict inequality holds when $0 < p_\theta < 1$.

*Part 3: Attention Mechanism Propagates Information.* The SDSA output contains terms directly utilizing inactivated neurons:

$$S_{enh} \supseteq \begin{cases} W_{\mathcal{A}} \odot K_{\mathcal{A}} & \text{(activated-inactivated interaction)} \\ W_{\mathcal{I}} \odot K_{\mathcal{A}} & \text{(direct inactivated information)} \\ W_{\mathcal{A}} \odot K_{\mathcal{I}} & \text{(direct inactivated information)} \\ W_{\mathcal{I}} \odot K_{\mathcal{I}} & \text{(direct inactivated information)} \end{cases}$$

where $S_{enh}$ incorporates interactions from inactivated neurons, the support set expands to $\mathcal{S}_{enh}$, strictly containing $\mathcal{S}_{\text{SA}}$. By the data processing inequality and the information chain rule:

$$I(X; S_{\text{enh}}) = I(X; Q_{\mathcal{A}}, Q_{\mathcal{I}}, K_{\mathcal{A}}, K_{\mathcal{I}})$$
$$= \sum_{i,j} I(X_{ij}; Q_{\mathcal{A}}^{ij}, Q_{\mathcal{I}}^{ij}, K_{\mathcal{A}}^{ij}, K_{\mathcal{I}}^{ij})$$
$$> \sum_{i,j} I(X_{ij}; Q_{\mathcal{A}}^{ij}, K_{\mathcal{A}}^{ij})$$
$$= I(X; S_{SA})$$

where the strict inequality comes from Part 2 when $\exists$ inactivated neurons with $0 < X_{ij} < \theta$.

The conditional entropy result follows from the mutual information identity:

$$H(X|S) = H(X) - I(X; S)$$
$$H(X|S_{enh}) = H(X) - I(X; S_{enh})$$
$$< H(X) - I(X; S_{SA})$$
$$= H(X|S_{SA})$$

completing the proof.

## C  PROOF OF THE PROPOSITION 2

**Proposition 2. Gradient Dynamics and Stability of Distribution Alignment.** *Under the standard assumption that $\mathcal{O}_s \in [\epsilon, 1]$ and $\mathcal{O}_a \in [\epsilon, 1]$ for a small $\epsilon > 0$ (ensuring no log-domain singularity), the combined gradient effectively guides the optimization process to address the aforementioned challenges in practice:*

$$\nabla \mathcal{L}_{align} = \frac{\partial \mathcal{L}_{align}}{\partial \mathcal{O}_s} = \frac{1}{N} \left( \log \frac{\mathcal{O}_s}{\mathcal{O}_a} + 1 \right) + \frac{2}{N} (\mathcal{O}_s - \mathcal{O}_a),$$

where the gradient $\frac{1}{N} (\log \frac{\mathcal{O}_s}{\mathcal{O}_a} + 1)$ governs distributional alignment by measuring relative discrepancies in probabilistic structure. The gradient $\frac{2}{N} (\mathcal{O}_s - \mathcal{O}_a)$ enforces geometric fidelity by penalizing absolute pixel-wise errors.

*Further, the Frobenius norm of the gradient satisfies:*

$$||\nabla \mathcal{L}_{\text{align}}||_F \leq \frac{1}{N} \sqrt{N^2 (\log \epsilon + 1)^2} + \frac{2}{N} \sqrt{N^2 (1 - \epsilon)^2} = 3 - 2\epsilon + |\log \epsilon|,$$

This guarantees that gradients remain bounded for any matrix, which ensures convergence. Therefore, the $\nabla \mathcal{L}_{align}$ effectively minimizes the divergence between SDSA and ANN-based self-attention feature response distributions, thereby reducing spike-information distortion to enhance segmentation performance under cluttered or low-contrast conditions.

**Proof 2.** The proof establishes the boundedness of the gradient norm through two parts of analysis.

*Part 1: Gradient Component Derivation.* We first derive the gradient components of the distribution alignment loss term-wise.

For the KL-divergence component:

$$\frac{\partial}{\partial \mathcal{O}_s} \left[ \mathcal{O}_s \log \left( \frac{\mathcal{O}_s}{\mathcal{O}_a + \epsilon} \right) \right] = \frac{\partial}{\partial \mathcal{O}_s} \left[ \mathcal{O}_s \log \mathcal{O}_s - \mathcal{O}_s \log(\mathcal{O}_a + \epsilon) \right]$$
$$= \log \mathcal{O}_s + 1 - \log(\mathcal{O}_a + \epsilon)$$
$$= \log \frac{\mathcal{O}_s}{\mathcal{O}_a + \epsilon} + 1.$$

For the L2-distance component:

$$\frac{\partial}{\partial \mathcal{O}_s} \left[ \|\mathcal{O}_s - \mathcal{O}_a\|_2^2 \right] = 2(\mathcal{O}_s - \mathcal{O}_a).$$

The complete gradient combines both components with normalization:

$$\nabla \mathcal{L}_{\text{align}} = \frac{1}{N}\left(\log \frac{\mathcal{O}_s}{\mathcal{O}_a + \epsilon} + 1\right) + \frac{2}{N}(\mathcal{O}_s - \mathcal{O}_a).$$

*Part 2: Frobenius Norm Bound.* We establish element-wise bounds under the standard assumption that $\mathcal{O}_s \in [\epsilon, 1]$ and $\mathcal{O}_a \in [\epsilon, 1]$ for a small $\epsilon > 0$ (ensuring no log-domain singularity):

Since $\frac{\mathcal{O}_s}{\mathcal{O}_a} \in [\epsilon, \frac{1}{\epsilon}]$, we have:

$$\left|\log \frac{\mathcal{O}_s}{\mathcal{O}_a} + 1\right| \le |\log \epsilon| + 1, \quad |\mathcal{O}_s - \mathcal{O}_a| \le 1 - \epsilon,$$

These element-wise bounds extend to Frobenius norms:

$$\left\|\log \frac{\mathcal{O}_s}{\mathcal{O}_a} + 1\right\|_F \le N(|\log \epsilon| + 1), \quad \|\mathcal{O}_s - \mathcal{O}_a\|_F \le N(1 - \epsilon),$$

Applying the triangle inequality:

$$\|\nabla \mathcal{L}_{\text{align}}\|_F \le \frac{1}{N}\left\|\log \frac{\mathcal{O}_s}{\mathcal{O}_a} + 1\right\|_F + \frac{2}{N}\|\mathcal{O}_s - \mathcal{O}_a\|_F$$
$$\le (|\log \epsilon| + 1) + 2(1 - \epsilon)$$
$$= |\log \epsilon| + 3 - 2\epsilon.$$

completing the proof.

# D  OVERALL ORCHESTRATION OF MEDSPIKEFORMER

---

**Algorithm 1:** MedSpikeFormer training loop

---

1 **Input:** Training image $X$, Ground truth $Y$.

2 **Output:** Prediction mask $\hat{Y}$ /*$\hat{\text{Y}}$ is MedSpikeFormer' output mask.*/

3 **for** $t \Longleftarrow 1$ to 4 **do**  /*Feature Extraction.*/

4  $\quad f_p \leftarrow Conv(SpikeConv(f_e^{t-1}))$ /*This is operations of Patch Embedding. $f_e^{t-1}$ ($t = 1$) denotes the input $X$.*/

5  $\quad Q, K, V \leftarrow SNN(BN(W_Q f_p, W_K f_p, W_V f_p))$ /*$W_{Q/K/V}$ are learned weights, $BN(\cdot)$ is batch normalization, and $Q, K, V \in \{0, 1\}$*/

6  $\quad Q_{\mathcal{A}} \leftarrow Q, \ Q_{\mathcal{I}} \leftarrow 1 - Q, \ K_{\mathcal{A}} \leftarrow K, \ K_{\mathcal{I}} \leftarrow 1 - K$ /*Complementary Spike Decomposition.*/

7  $\quad S_{\text{enh}} \leftarrow \copyright\Big([Q_{\mathcal{A}}, Q_{\mathcal{I}}] \otimes [K_{\mathcal{A}}, K_{\mathcal{I}}]^T; [Q_{\mathcal{I}}, Q_{\mathcal{A}}] \otimes [K_{\mathcal{A}}, K_{\mathcal{I}}]^T\Big)$

8  $\quad \mathcal{O}_s \leftarrow \copyright\Big(\frac{[Q_{\mathcal{A}}, Q_{\mathcal{I}}] \otimes [K_{\mathcal{A}}, K_{\mathcal{I}}]^T}{\sqrt{C}} \otimes V; \frac{[Q_{\mathcal{I}}, Q_{\mathcal{A}}] \otimes [K_{\mathcal{A}}, K_{\mathcal{I}}]^T}{\sqrt{C}} \otimes V\Big)$ /*$\copyright(\cdot)$ is concatenation operation.*/

9  $\quad f_e^t \leftarrow SpikeConv(\mathcal{O}_s) + f_p$ /*$f_e^i$ is the output of SDQK-A of $t$-th layer.*/

10 **for** $t \Longleftarrow 4$ to 1 **do**  /*Feature Fusion.*/

11  $\quad$ **if** $t == 4$ **do**

12  $\quad\quad f_s^t \leftarrow SpikeConv(f_e^t)$

13  $\quad$ **else**

14  $\quad\quad f_s^t \leftarrow SpikeConv(f_e^t + f_s^{t+1})$

15  $\quad$ /*Loss Function $\mathcal{L}_{Seg}$: region-level and pixel-level supervision.*/

16  $\quad \hat{Y} = \sigma(\phi_s(f_s^1, \omega_s))$ /*Generating prediction $\hat{\text{Y}}$. $\sigma$ is the sigmoid function*/ /*$\phi_s(\cdot)$ is a map function. $\omega_s$ is learned weights.*/

17  $\quad \mathcal{L}_{Seg}(\hat{Y}, Y) \leftarrow \underbrace{1 - \frac{2\sum_i \hat{Y}_i Y_i + \epsilon}{\sum_i \hat{Y}_i^2 + \sum_i Y_i^2 + \epsilon}}_{\mathcal{L}_{Dice}} + \underbrace{-\sum_i \Big[Y_i \log(\hat{Y}_i) + (1 - Y_i)\log(1 - \hat{Y}_i)\Big]}_{\mathcal{L}_{BCE}}$

---

# E  EXPERIMENT

## E.1  DATASETS AND EXPERIMENT DETAILS

**Datasets.** Our method is evaluated on 5 different modalities of public datasets, including ISIC2018, Kvasir, BUSI, Moun-Seg, and COVID-19.

The ISIC2018 is a relative large dataset, which contains 2594 skin lesion images for the task of skin cancer detection, with 2076 images used for training and 518 images for testing.

The Kvasir dataset focuses on pixel-level segmentation of colorectal polyps and includes 1,000 endoscopic images, with 800 images used for the training set and 200 images for the test set.

The BUSI is a breast ultrasound imaging dataset that is categorized into three classes: normal, benign, and malignant, comprising a total of 780 images, with 624 images used for training and 156 images for testing.

The Monu-Seg is a medical imaging dataset for cell nucleus segmentation, containing 74 images, with 59 images used for training and 15 images for testing.

The COVID-19 dataset contains 894 images for the segmentation of CT images of lung infection regions, with 716 images used for training and 178 images for testing.

**Implementation Details.** We use PyTorch on NVIDIA TITAN RTX GPU. The optimization is AdamW, and the learning rate scheduler is CosineAnnealingLR. We resize input images to 256 $\times$ 256. To enhance model robustness, we use horizontal flipping, vertical rotation, and rotation operations for data augmentation. The training epoch is set as 200 and the initial learning rate is set to be 1e-4. Plus, the batchsize is set as 12, and the seed is set as 41.

## E.2  STATISTICAL SIGNIFICANCE

To assess statistical significance, we conduct paired t-tests comparing our method with other SOTA approaches across five datasets. As shown in Table 1, our method consistently achieves statistically significant improvements ($p < 0.05$) over most baselines, validating the robustness of our performance gains.

Table 1: Paired t-test $p$-values comparing our method with other SOTAs.

| Model vs. Ours | $p$-value |
| --- | --- |
| U-Net | 0.0829 |
| UCTransNet | 0.1855 |
| D-LKA | 0.0791 |
| EGE-UNet | 0.0266 |
| SAM-Med2D | 0.0144 |
| SDSA | 0.0349 |
| MLW-Net | 0.0255 |
| UltraLight VM-UNet | 0.0122 |
| MFMSA | 0.0122 |
| VPTTA | 0.0103 |
| EMCAD | 0.0218 |
| QKFormer | 0.0120 |
| STDV3 | 0.1384 |
| FSTA-SNN | 0.0235 |

## E.3  COMPARISON WITH STATE-OF-THE-ARTS

As shown in Table 2, our method has obvious advantages. Plus, Figure 1 show the superior capability of our method in image segmentation tasks Notably, as shown in Fig 2, on the ISIC2018 dataset, the mIoU curve of our method maintains a stable upward trend when trained to 80 epochs. It is worth noting that on the Kvasir dataset, our method continues to improve performance and significantly surpasses other methods at 100 epochs. On the Monu-Seg dataset, other methods reach

performance saturation at 125 epochs, while our method can still continue to optimize. Comprehensive experimental results show that our method has obvious advantages in both model convergence and stability in medical image segmentation tasks.

Table 2: Performance comparison with 14 SOTA methods on ISIC2018, Kvasir, BUSI, COVID-19 and Monu-Seg datasets.

| Dataset | Metric | U-Net MICCAI 2015 | UCTrans Net AAAI 2022 | D-LKA WACV 2023 | EGE-UNet MICCAI 2023 | SAM-Med2D arXiv 2023 | SDSA NeurIPS 2023 | MLW-Net CVPR 2023 | UltraLight VM-UNet arXiv 2024 | MFMSA CVPR 2024 | VPTTA CVPR 2024 | EMCAD CVPR 2024 | QKFormer NeurIPS 2024 | STDV3 TPAMI 2025 | FSTA -SNN AAAI 2025 | Our 2025 |
|---|---|---|---|---|---|---|---|---|---|---|---|---|---|---|---|---|
| ISIC 2018 | mIoU↑ | 0.8004 | 0.8185 | 0.8033 | 0.8108 | 0.7383 | 0.7853 | 0.7650 | 0.8110 | 0.8163 | 0.7842 | 0.8071 | 0.7706 | **0.8303** | 0.6403 | **0.8550** |
| | DSC↑ | 0.8891 | 0.9002 | 0.8909 | 0.8955 | 0.8494 | 0.8579 | 0.8613 | 0.8956 | 0.8988 | 0.8790 | 0.8932 | 0.8614 | **0.8965** | 0.7400 | **0.9081** |
| | Acc↑ | 0.9513 | 0.9565 | 0.9514 | 0.9535 | 0.9397 | 0.9572 | 0.9404 | 0.9527 | 0.9549 | 0.9467 | 0.9531 | 0.9475 | **0.9666** | 0.9110 | **0.9812** |
| | Spe↑ | 0.9738 | 0.9794 | 0.9742 | 0.9721 | **0.9970** | 0.9816 | 0.9768 | 0.9662 | 0.9724 | 0.9699 | 0.9750 | 0.9718 | 0.9812 | 0.9728 | **0.9896** |
| | Sen↑ | 0.8730 | 0.8769 | 0.8738 | 0.8891 | 0.7457 | 0.8432 | 0.8161 | **0.9061** | 0.8943 | 0.8659 | 0.8769 | 0.8703 | 0.8951 | 0.7810 | **0.9177** |
| Kvasir | mIoU↑ | 0.7330 | 0.7670 | 0.7212 | 0.5604 | 0.5911 | 0.7970 | 0.6636 | 0.6100 | 0.7625 | 0.5164 | 0.7173 | 0.7658 | **0.8215** | 0.6313 | **0.8534** |
| | DSC↑ | 0.8459 | 0.8681 | 0.8380 | 0.7182 | 0.7430 | 0.8726 | 0.7977 | 0.7577 | 0.8652 | 0.6811 | 0.8353 | 0.8521 | **0.8915** | 0.7642 | **0.9093** |
| | Acc↑ | 0.9253 | 0.9579 | 0.9495 | 0.9108 | 0.9324 | 0.9741 | 0.9378 | 0.9190 | 0.9568 | 0.9076 | 0.9478 | 0.9662 | **0.9722** | 0.9264 | **0.9804** |
| | Spe↑ | 0.9786 | 0.9767 | 0.9760 | 0.9499 | **0.9964** | 0.9805 | 0.9715 | 0.9441 | 0.9750 | 0.9639 | 0.9716 | 0.9844 | 0.9813 | 0.9815 | **0.9877** |
| | Sen↑ | 0.8145 | 0.8600 | 0.8115 | 0.7068 | 0.6020 | 0.8764 | 0.7620 | 0.8617 | 0.8617 | 0.6137 | 0.8234 | 0.8622 | **0.9177** | 0.7246 | **0.9259** |
| Moun -Seg | mIoU↑ | 0.6784 | **0.6890** | 0.6300 | 0.5009 | 0.2699 | 0.6554 | 0.6535 | 0.5600 | 0.6111 | 0.4151 | 0.5603 | 0.6003 | 0.6696 | 0.6495 | **0.7014** |
| | DSC↑ | 0.8084 | **0.8159** | 0.7730 | 0.6674 | 0.4250 | 0.7883 | 0.7904 | 0.7180 | 0.7586 | 0.5867 | 0.7182 | 0.7475 | 0.8008 | 0.5834 | **0.8238** |
| | Acc↑ | 0.9348 | 0.9433 | 0.9284 | 0.8873 | 0.9433 | 0.9414 | 0.9332 | 0.9094 | 0.9338 | 0.8662 | 0.9263 | 0.9335 | **0.9467** | 0.9095 | **0.9518** |
| | Spe↑ | 0.9493 | **0.9663** | 0.9532 | 0.9149 | **0.9977** | 0.9515 | 0.9540 | 0.9386 | 0.9493 | 0.9188 | 0.9522 | 0.9497 | 0.9613 | 0.9463 | 0.9615 |
| | Sen↑ | **0.8587** | 0.8164 | 0.7920 | 0.7351 | 0.2774 | 0.8296 | 0.8187 | 0.7489 | 0.8261 | 0.5911 | 0.7461 | 0.7718 | 0.8063 | 0.5658 | **0.8598** |
| COVID -19 | mIoU↑ | 0.3605 | 0.3971 | 0.3098 | 0.3912 | 0.4025 | 0.5062 | 0.4295 | 0.5532 | **0.6262** | 0.4591 | 0.4120 | 0.4701 | 0.5974 | 0.4902 | **0.7138** |
| | DSC↑ | 0.5300 | 0.5684 | 0.4730 | 0.5624 | 0.5739 | 0.6261 | 0.6009 | 0.7123 | **0.7201** | 0.6293 | 0.5835 | 0.6357 | 0.6933 | 0.6517 | **0.8009** |
| | Acc↑ | 0.9784 | 0.9804 | 0.9766 | 0.9805 | 0.9856 | 0.9782 | 0.9808 | 0.9867 | **0.9901** | 0.9859 | 0.9790 | 0.9715 | 0.9286 | 0.9713 | **0.9565** |
| | Spe↑ | 0.9881 | 0.9902 | 0.9888 | 0.9909 | 0.9981 | 0.9891 | 0.9893 | 0.9933 | **0.9966** | **0.9968** | 0.9872 | 0.9938 | 0.9596 | 0.9915 | 0.9787 |
| | Sen↑ | 0.5488 | 0.5626 | 0.4574 | 0.5397 | 0.4359 | **0.7922** | 0.6221 | 0.7092 | 0.7139 | 0.5203 | 0.6334 | 0.6026 | 0.7406 | 0.6749 | **0.8401** |
| BUSI | mIoU↑ | 0.4775 | **0.5870** | 0.4969 | 0.5103 | 0.4770 | 0.5115 | 0.4811 | 0.4743 | 0.5771 | 0.4420 | 0.4620 | 0.4310 | 0.2854 | 0.5582 | **0.6006** |
| | DSC↑ | 0.6463 | **0.7382** | 0.6639 | 0.6758 | 0.6459 | 0.5917 | 0.6496 | 0.6434 | **0.7318** | 0.6131 | 0.6320 | 0.6041 | 0.3774 | 0.7165 | 0.6676 |
| | Acc↑ | 0.9605 | 0.9637 | 0.9502 | 0.9551 | 0.9616 | 0.9571 | 0.9491 | 0.9462 | 0.9629 | 0.9426 | 0.9412 | 0.9377 | 0.9362 | **0.9652** | **0.9759** |
| | Spe↑ | 0.9906 | 0.9852 | 0.9732 | 0.9810 | **0.9963** | 0.9735 | 0.9742 | 0.9696 | 0.9853 | 0.9689 | 0.9620 | 0.9750 | 0.9898 | 0.9874 | **0.9906** |
| | Sen↑ | 0.5384 | **0.6955** | 0.6632 | 0.6313 | 0.5003 | 0.6404 | 0.6360 | 0.6540 | **0.6828** | 0.6136 | 0.6811 | 0.5871 | 0.3750 | 0.6558 | 0.6667 |
| Params (M) ↓ | | 14.7518 | 66.2424 | 22.8401 | 0.0458 | - | 13.5588 | 94.9794 | 0.0376 | 31.2192 | 22.0224 | 26.7643 | 16.9599 | 25.5286 | 45.6777 | 1.7369 |
| GFLOPs ↓ | | 32.8948 | 30.9839 | 16.8894 | 0.0072 | - | 28.5805 | 108.0758 | 0.0602 | 9.9752 | 40.0514 | 5.5960 | 32.8858 | 12.3349 | 20.6515 | 19.0929 |

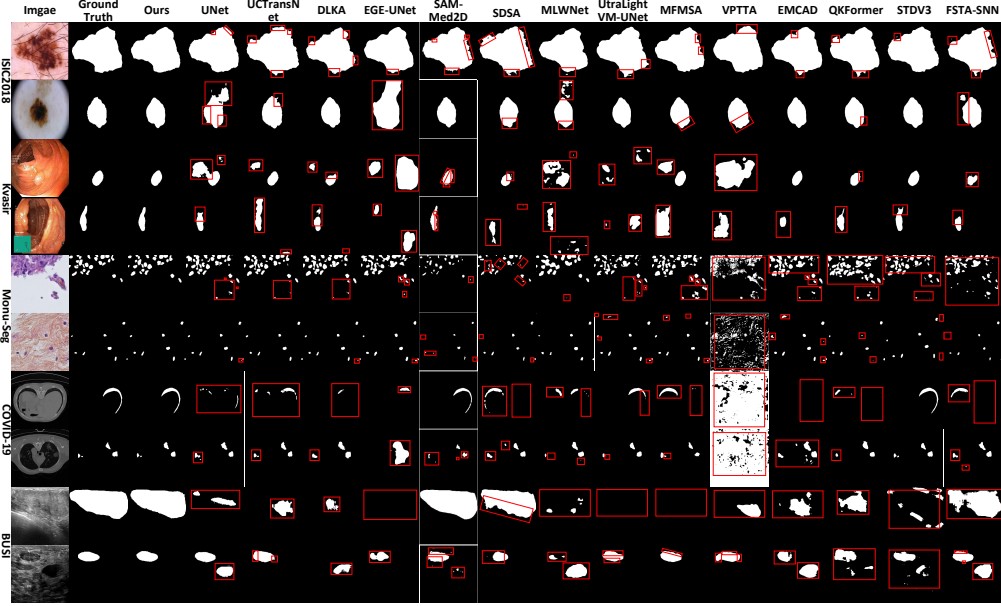

Figure 1: We compare our method with 14 state-of-the-art methods. The red box indicates the area of incorrect predictions.

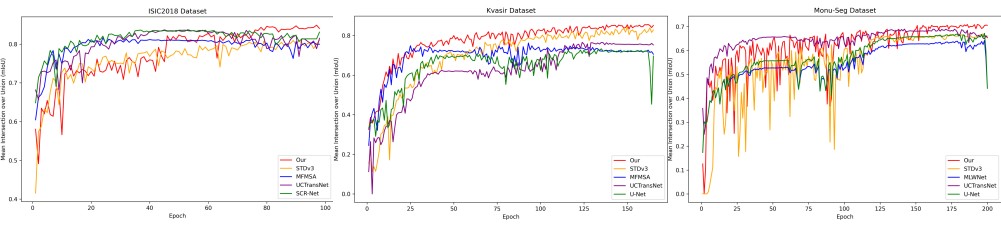

Figure 2: The Mean Intersection over Union (mIoU) curves.

### E.4 COMPUTATIONAL ENERGY CONSUMPTION

Though not lightweight-oriented, MedSpikeFormer keeps low overhead. It outperforms other spike-based methods, such as STDV3 and QKFromer. These results show that our method is hardware-friendly.

Table 3: Computational energy consumption comparison.

| Model | Params(M) | Gflops | Power(mJ) |
|---|---|---|---|
| Ours | 1.7369 | 19.0929 | 1.909 |
| STDV3 | 25.5286 | 12.3349 | 7.4 |
| QKFormer | 16.9599 | 32.8858 | 15.5709 |

### E.5 ABLATION STUDIES

We validate the contribution of our method to segmentation performance on 5 datasets using the mIoU metric. Specifically, we conduct extensive experimental evaluations to answer the following questions:

Q1: How critical is SDSA to segmentation performance? As shown in Fig. 3, when removing SDSA from MedSpikeFormer **(w/o SDSA)** on the ISIC2018 dataset, the model struggles to suppress interference from non-salient objects, leading to decreased accuracy in detecting segmentation boundaries. On the Monu-Seg dataset, where blurry backgrounds coexist with multiple small objects, the variant **(w/o SDSA)** fails to reliably detect individual small objects. Other datasets exhibit similar performance degradation patterns. In contrast, our proposed method demonstrates superior segmentation performance in scenarios with misleading co-occurrence between sali ent and non-salient objects, effectively addressing these challenges. These results fully validate the effectiveness of SDSA. Plus, we further provide heatmaps to show SDSA's effectiveness, as shown in Figure 4.

Q2: Does the ANN-based self-attention (SA) module significantly affect performance? Yes. As shown in Fig. 3, when removing ANN-based self-attention (w/o SA), the performance of model significantly descends. These results fully validate its effectiveness for segmentation performance.

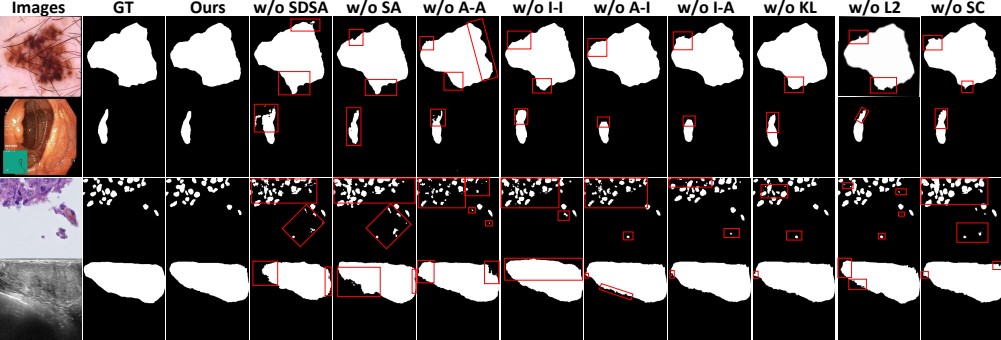

Figure 3: Ablation Visualization comparison. The red box indicates the area of incorrect predictions.

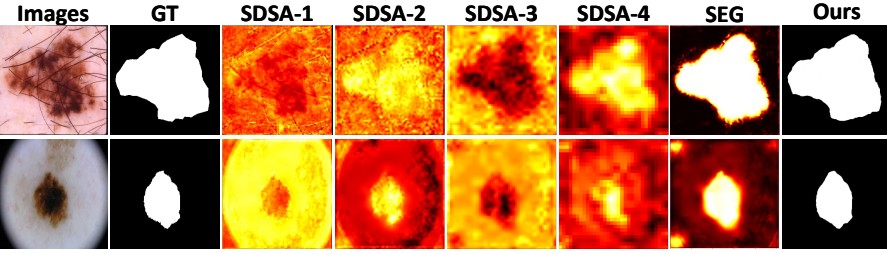

Figure 4: Ablation Visualization comparison. The red box indicates the area of incorrect predictions.

Q3: Does the Spike Convolution (SC) module improve performance? Yes. Similarly, as shown in Fig. 3, removing the Spike Convolution (w/o SC) leads to notable performance degradation. These results fully validate its effectiveness for segmentation performance.

Q4: Do all four types of spike interactions in SDSA matter? Yes. Fig. 3 shows that removing any of the four interaction types (w/o A-A, w/o A-I, w/o I-A, and w/o I-I) leads to performance drop. In particular, both A-A and I-I interactions demonstrate a significant impact on the segmentation performance. This confirms that all interaction pairs contribute to comprehensive information modeling.

Q5: Does the distribution alignment loss improve segmentation performance? Yes. The distribution alignment loss consists of both KL divergence and $L_2$ loss. As shown in Table. 4, removing either loss component leads to a drop in segmentation performance, with the $L_2$ loss having a particularly significant impact. This result confirms the effectiveness of the distribution alignment loss. Plus, as shown in Fig. 3, removing either loss component leads to a drop in segmentation performance, with the $L_2$ loss having a particularly significant impact. This result confirms the effectiveness of the distribution alignment loss.

Table 4: Ablation on the distribution loss (mIoU).

| Model Variant | ISIC2018 | Kvasir | Monu-Seg | COVID-19 | BUSI |
|---|---|---|---|---|---|
| Ours | **0.8550** | **0.8534** | **0.7014** | **0.7138** | **0.6006** |
| w/o KL | 0.7743 | 0.7782 | 0.4988 | 0.3988 | 0.5108 |
| w/o $L_2$ | 0.7743 | 0.7782 | 0.4988 | 0.3988 | 0.5108 |

Q6: Is the timestep $D$ in Our method important? Yes. Table 5 shows that both under-quantized ($D = 1, 2$) and over-quantized ($D = 6$) configurations reduce segmentation performance. Our design with $D = 4$ yields optimal results.

Table 5: Ablation on timestep $D$ in MedSpikeFormer (mIoU).

| Time Step | ISIC2018 | Kvasir | Monu-Seg | COVID-19 | BUSI |
|---|---|---|---|---|---|
| (D=4) Ours | **0.8550** | **0.8534** | **0.7014** | **0.7138** | **0.6006** |
| D = 1 | 0.7545 | 0.7082 | 0.4444 | 0.4275 | 0.4516 |
| D = 2 | 0.7995 | 0.7314 | 0.4536 | 0.4387 | 0.3780 |
| D = 6 | 0.8158 | 0.8174 | 0.7039 | 0.6761 | 0.5332 |

## F    OBSERVATIONS AND DISCUSSION

We highlight two key observations:

**Salient object detection in scenarios with blurred edges.** Our method performs well in scenarios with blurred boundaries, successfully identifying the salient objects. However, there are still minor differences compared to the ground truth. This suggests that the model's ability to precisely capture object boundaries requires further refinement. Therefore, we plan to explore advanced edge-aware techniques to enhance the model's segmentation performance.

**Multiple object detection in the co-occurrence scenario of salient and non-salient objects.** Our method performs favorably on medical images containing multiple small objects, successfully detecting each of the small objects. Nonetheless, there remain slight mismatches with the ground truth. This indicates that the model still has room for improvement in accurately segmenting multiple small objects. Therefore, we plan to develop efficient spike convolution mechanisms to better capture fine-grained details.