# OpenReview forum: "MEDSPIKEFORMER: All Neurons Matter for Medical Image Segmentation"
_ICLR.cc/2026/Conference — ICLR 2026 Conference Withdrawn Submission_

### Official Review · Reviewer_hz6Z · 2025-10-15

**Soundness:** 2
**Presentation:** 2
**Contribution:** 3
**Rating:** 4
**Confidence:** 4

**Summary:**

To address the issues of spiking self-attention in medical image segmentation—where it only relies on activated neurons and suffers from distribution distortion due to binary spike representation, the researchers propose MedSpikeFormer, a spiking Transformer adhering to the principle that "all neurons matter". It explicitly models four types of neuronal interactions—A–A, A–I, I–A, and I–I—via SDSA to recover rich contextual dependencies. Additionally, a distribution alignment loss is introduced to enhance the discriminability of spatial features. Experiments on five medical segmentation benchmark datasets show that this method consistently outperforms 14 SOTA methods, achieving an mIoU improvement of up to +2.4% on ISIC2018 and +8.7% on the COVID-19 dataset. These results confirm that leveraging both activated and inactivated neurons is crucial for robust spike-driven medical image segmentation.

**Strengths:**

1 The experimental section includes statistical significance tests.

2 It conducts comparisons of SNN-based models at the power consumption level.

3 It expands the application of SNNs in the field of medical image segmentation.

**Weaknesses:**

1 Does MedSpikeFormer have an advantage in inference speed compared to other lightweight methods?

2 All datasets selected in the experiments are for binary classification tasks. The authors did not validate the model’s effectiveness on multi-class segmentation tasks (e.g., Synapse dataset), which limits the scope of the comparative experiments in this paper.

3 In Table 3, when the A–I interaction is removed, the mIoU decreases by only 0.16% (on ISIC2018), while removing the I–I interaction leads to a 2.06% decrease in mIoU. However, the authors did not analyze the reasons for this discrepancy—for example, whether the A–I interaction carries less information in medical images.

4 In Table 3, why do the results of w/o KL and w/o L2 show exactly the same performance? This requires more in-depth analysis.

5 The transition word used throughout the paper seems to be "Plus", but the phrase "Plus, xxx" is relatively uncommon.

6 There may be potential unfairness in the comparative experiments. The authors claim that all models were trained and evaluated under the same experimental settings, but the optimal training parameter settings for MedSpikeFormer may not necessarily be optimal for other methods.

If the authors can address the above concerns, I am willing to raise the score.

**Questions:**

see weakness

---

### Official Review · Reviewer_VCLo · 2025-10-26

**Soundness:** 2
**Presentation:** 3
**Contribution:** 2
**Rating:** 4
**Confidence:** 4

**Summary:**

This paper proposes MedSpikeFormer, a spiking transformer for medical image segmentation. It introduces Spike-based Decomposed Self-Attention (SDSA) to explicitly model interactions between activated and inactivated neurons, recovering contextual dependencies overlooked by conventional SSA. A distribution alignment loss is further designed to align SDSA with ANN-based self-attention, enhancing spatial discriminability while preserving spike efficiency.

**Strengths:**

1. By enabling a more effective transfer of contextual information from ANN-based attention to SNNs, the work broadens the applicability of spiking transformers to complex medical scenarios. The proposed framework provides new insight into bridging biological plausibility and transformer-like representation learning.
2. The overall architecture is intuitively organized, and the figures effectively illustrate how SDSA expands spike attention scope to include interactions among activated and inactivated neurons.
3. Relative to prior SNN-based segmentation work, the paper contributes two conceptually novel elements including SDSA and a distribution-alignment scheme between SDSA and ANN-based self-attention. Together, these components effectively bridge the gap between event-driven and continuous attention, marking a meaningful step toward unifying ANN and SNN representations.

**Weaknesses:**

1. The distribution-alignment loss is not defined in a fully verifiable manner. Two essential aspects remain underspecified: It is not stated whether \hat{O}_{s} and \hat{O}_{a} are normalized into valid probability distributions. Without an explicit normalization map (e.g., a SoftMax with temperature) and a declared normalization axis (channel / spatial / token), the KL term is not mathematically valid. Moreover, the alignment granularity (per-pixel, head, or layer) is not described, making the formulation difficult to reproduce or verify.
2. Results are reported without standard deviations or confidence intervals. Please provide mean ± std over ≥3 random seeds for all key metrics.
3. The reported computational efficiency values are inconsistent. In the Section 4.2, the model is described as having 2.9 M parameters and 26.87 GFLOPs, while Table 1 lists 1.7369 M parameters and 19.0929 GFLOPs for the same configuration.
Such numerical discrepancies undermine the credibility and reproducibility of the results.
4. In Table 3, the results for w/o KL and w/o L2 are the same, making it impossible to observe the individual contribution of each loss component. This contradicts the claim that removing either term affects segmentation performance.

**Questions:**

1. Is the ANN-based self-attention signal realistically available during training in real medical scenarios, or does it rely on additional modules or pre-trained networks that may limit scalability?
2. Can the proposed SDSA module be integrated as a plug-in within existing SNN or ANN transformer frameworks, or is it tightly coupled to the specific MedSpikeFormer architecture?
3. Would the proposed distribution alignment still be beneficial in non-medical domains (dynamic vision sensors, general object segmentation), or is the improvement domain-specific to medical segmentation?
4. If the ANN-based self-attention signal used as a teacher for distribution alignment were removed, would the model still maintain competitive performance purely as an attention-augmented SNN?
5. For the reported GFLOPs in SNNs, did the authors compute them based on a time-step–equivalent FLOPs estimation?
6. Could the authors make all relevant runnable code, including ablation experiments, test results, and the different datasets used in this study, publicly available?

---

### Official Review · Reviewer_5dWC · 2025-10-29

**Soundness:** 2
**Presentation:** 3
**Contribution:** 2
**Rating:** 4
**Confidence:** 3

**Summary:**

This paper proposes MedSpikeFormer, a spiking transformer framework for medical image segmentation. MedSpikeFormer introduces the Spike-based Decomposed Self-Attention (SDSA) to consider inactivated neurons, thereby recovering lost contextual dependencies and a Distribution Alignment Loss to minimize the distribution gap between SDSA and ANN-based SA, preserving spike sparsity while boosting spatial discriminability. Although effective, this article still has some deficiencies in terms of novelty, motivation, experiment comparisons and Interpretability. In my view, this paper is marginally below the acceptance bar of ICLR.

**Strengths:**

1. By decomposing spikes into activated and inactivated components, SDSA fills a critical gap in existing SSA (which ignores inactivated neurons).

2. Comprehensive experiments show that the proposed method Outperforms 14 SOTA methods with significant gains on challenging datasets (e.g., +8.06% mIoU over MFMSA on COVID-19).

**Weaknesses:**

1. The effectiveness of the motivation in SDSA needs further verification because it involves multiple attention blocks. The gain may be merely due to the increase in the number of attention modules and parameters, rather than the development of inactivated neurons.

2. Distribution alignment loss seems to be just a combination of KL and L2 losses.  At the same time, there is a lack of comparison with other distributed loss functions, which reduces the innovativeness of this module.

3. There is a lack of performance comparison with existing spiking self-attention methods, such as those mentioned in the Introduction.

**Questions:**

1. Does the model parameter include the parameters of the SA module?

2. Why are the parameters of your model in Table 1 (1.7369) different from those in Table 2 (2.8745)?

---

### Official Review · Reviewer_kdK9 · 2025-10-31

**Soundness:** 2
**Presentation:** 2
**Contribution:** 2
**Rating:** 2
**Confidence:** 4

**Summary:**

This paper proposes MedSpikeFormer, a spiking transformer architecture for medical image segmentation.

**Strengths:**

1. The paper identifies a limitation of existing spiking self-attention mechanisms.
2. The paper provides extensive ablations validating each component (SDSA, each interaction type, distribution alignment loss components).

**Weaknesses:**

1. The notation [QA, QI] ⊗ [KA, KI]^T is unclear. Does this represent concatenation followed by matrix multiplication? If so, the dimensional analysis is missing. How exactly are the four interaction types computed and combined?
2. The paper claims to preserve binary spike properties while simultaneously forcing SDSA outputs to match continuous-valued ANN attention distributions through Lalign. This fundamentally undermines the event-driven, sparse nature that makes SNNs attractive.
3. The gradient bound ||∇Lalign||F ≤ 3 - 2ε - log ε is stated, but its practical significance is unclear. The claim that this "guarantees convergence" is too strong, a bounded gradient is necessary but not sufficient for convergence. Standard convergence analysis (e.g., Lipschitz continuity, convexity) is missing.
4. The distribution alignment approach is essentially knowledge distillation from ANN to SNN, which is well-established. The combination with SDSA doesn't yield significant theoretical insight.
5. The paper's central motivation is energy efficiency of SNNs, yet no actual energy measurements on neuromorphic hardware are provided. FLOPs comparisons with ANNs are misleading since SNNs have fundamentally different computational characteristics.
6. "All neurons matter" is misleading: Standard transformers already consider all positions through attention. The framing as a novel insight is overstated, the paper is just explicitly considering the inverse of a binary mask.

**Questions:**

1. Can you provide explicit matrix dimensions for Equations 8-9 and clarify how the four interaction types are computed separately vs. jointly?
2. How does the distribution alignment loss affect spike rates? Do you measure and report actual sparsity metrics?
3. How does performance vary with timestep T? What is the computational cost of multiple timesteps?
4. In Table 3, removing A-I and I-A shows minimal impact. How do you explain this given your claims about their importance?

---

### Note · Authors · 2025-11-19

I have read and agree with the venue's withdrawal policy on behalf of myself and my co-authors.